# Inflammatory Bowel Disease in Adult HIV-Infected Patients-Is Sexually Transmitted Infections Misdiagnosis Possible?

**DOI:** 10.3390/jcm11185324

**Published:** 2022-09-10

**Authors:** Ewa Siwak, Magdalena M. Suchacz, Iwona Cielniak, Joanna Kubicka, Ewa Firląg-Burkacka, Alicja Wiercińska-Drapało

**Affiliations:** 1Department of Infectious and Tropical Diseases and Hepatology, Medical University of Warsaw, 02-091 Warsaw, Poland; 2HIV Out-Patient Clinic, Hospital for Infectious Diseases, 02-091 Warsaw, Poland; 3Hospital for Infectious Diseases, 02-091 Warsaw, Poland

**Keywords:** IBD, STIs, HIV, misdiagnosis

## Abstract

Background. The aim of our study was to describe 50 cases of inflammatory bowel disease (IBD) and HIV co-existence that are under medical supervision in Warsaw. Methods. This was a retrospective descriptive study. Fifty HIV-infected patients, diagnosed with IBD during the years 2001–2019, were identified. IBD was diagnosed endoscopically and then confirmed by biopsy. All data was obtained from medical records. Results. All studied patients were male with a median age of 33 years old (range 20–58 years). All, except one, were men who have sex with men (MSM). The median CD4 cell count was 482 cells/µL (range 165–1073 cells/µL). Crohn’s disease (CD) was diagnosed in 7 patients (14%), ulcerative colitis (UC) in 41 patients (82%), and 2 patients (4%) had indeterminate colitis. Forty-nine patients (98%) reported a history of unprotected receptive anal intercourse and different sexual transmitted infections (STIs). Only in 10 patients (20%) were one or more IBD relapses observed. Conclusions. We recommend HIV testing for every MSM with IBD suspicion. Moreover, STIs testing should be performed in every IBD patient with colorectal inflammation, using molecular and serological methods. Persons who reported unprotected receptive anal intercourse seem to have the biggest risk of STI-associated proctitis or proctocolitis mimicking IBD.

## 1. Introduction

Inflammatory bowel disease (IBD) is an idiopathic chronic relapsing inflammatory disease of the intestine and is classified into two major subtypes: ulcerative colitis (UC), which primarily affects the colon, and Crohn’s disease (CD), which affects different parts of the gastrointestinal tract. IBD is characterized by relapses and remissions and requires chronic treatment. Both subtypes are characterized by diarrhea, abdominal pain, fatigue and weight loss. However, in spite of many clinical similarities, UC and CD are differentiated based upon their histopathological picture and the primary affected anatomical location. UC is characterized as superficial mucosal inflammation found continuously extending usually from the rectum throughout the length of the colon. Contrarily, CD can potentially affect any part of the gastrointestinal tract [1,2]. Moreover, the IBD clinical picture may also depend on other factors such as age, microbiome or dietary habits [3]. The causative mechanisms of IBD still remain unknown. However, it seems that its pathophysiology is a complex interaction between the adaptive and innate immune systems, the microbiome and genetic and environmental factors [4]. One of the major hypotheses that could explain IBD etiopathogenesis is persistent inflammation triggered by an unknown environmental antigen on a genetically susceptible host [5]. CD is a T cell mediated disease, and UC is an antibody mediated disease; however, in both diseases, the final effector pathways are common. It has been shown that CD4+ T lymphocytes associated with the gut-associated lymphoid tissue (GALT) play an important role in IBD pathogenesis [6].

GALT contains almost 60% of all CD4+ T cells in the body. Approximately 95% of them are immune memory T cells, which are the most important site of human immunodeficiency virus (HIV) replication in every phase of infection [7,8]. Unlike the peripheral lymph nodes, spleen and blood, there is no lymphocyte subpopulation heterogeneity in the GALT. Therefore, the largest decrease in the number of T cells during HIV infection occurs in GALT. Recovery of the immune system and increasing CD4 cell count in effective antiretroviral (ARV) treatment, seen in the periphery, is not associated with the complete recovery of CD4 cells in GALT. Consequently, the intestinal mucosal tissues remain one of the most important sites of persistent HIV infection in humans [9,10,11]. As a result, IBD and HIV infection can lead to a dysfunction of the intestinal mucosal barrier and increased mucosal permeability. Moreover, HIV-infected patients may additionally present different opportunistic infections caused by viruses (cytomegalovirus—CMV, herpes simplex virus—HSV), bacteria (Mycobacterium avium complex—MAC, *Clostridium* spp., *Salmonella* spp., *Shigella* spp., *Campylobacter* spp.), fungi (*Candida* spp.) and parasites (*Microsporidium* spp., *Cryptosporidium* spp.) or cancers, such as Kaposi sarcoma and lymphoma, that may also damage the intestinal mucosa [12,13].

Both IBD and HIV infection may coexist in the same person; however, this situation is rather uncommon. IBD and HIV infection are characterized by microbial translocation and chronic systemic inflammation. However, they differ in terms of the severity of gut damage and the mechanisms of immune cell homeostasis [14]. As a result, the relationship between HIV and IBD course is complex and still unknown.

Sexually transmitted infections (STIs) are spread by sexual contact, including vaginal, anal and oral sex. The most common pathogens caused STIs are Treponema pallidum, Neisseria gonorrhoeae, Chlamydia trachomatis, Trichomonas vaginalis, hepatitis B virus (HBV), herpes simplex virus (HSV), HIV and human papillomavirus (HPV). In 2020, WHO estimated 374 million new infections: chlamydia (129 million), gonorrhoea (82 million), syphilis (7.1 million) and trichomoniasis (156 million) [15]. It has been shown that HIV and other STI incidence rates are higher among sexual and gender minority populations, such as men who have sex with men (MSM) or transgender persons, especially among those aged < 30 years and among substance-using individuals [16]. Several STIs, such as Treponema pallidum, Neisseria gonorrhoeae, Chlamydia trachomatis, Trichomonas vaginalis, Mycoplasma genitalium and HSV may cause proctitis, with such symptoms as diarrhea, rectal bleeding, tenesmus or anal discharge. These symptoms may mimic IBD, because even endoscopy and histopathological examination may be similar to IBD with mucosal inflammation, ulceration, granuloma formation, colorectal fistulas, etc. [17,18].

The aim of our study was to describe 50 cases of IBD/HIV co-existence under medical supervision in our outpatient and inpatient settings and searching the factors suggesting possible STI proctitis/IBD misdiagnosis in this uncommon population.

## 2. Materials and Methods

This was a retrospective descriptive study. All 50 HIV-infected patients who participated in this study were on combined antiretroviral therapy (cART) and were under the uninterrupted care of the HIV Out-Patient Clinic at the Hospital for Infectious Diseases in Warsaw and of the Department of Infectious and Tropical Diseases and Hepatology of the Medical University of Warsaw in Poland in the years 2001–2019. The inclusion criteria were confirmed HIV infection, confirmed IBD diagnosis and age > 18 years old. IBD was diagnosed based on typical clinical symptoms, laboratory findings, endoscopic picture (in colonoscopy or sigmoidoscopy) and histopathological examination. The Truelove and Witts criteria have been used to define acute severe colitis. The IBD diagnosis and treatment were conducted by gastroenterologists who occasionally consulted with HIV specialist. IBD was diagnosed before HIV infection in 9 subjects, concomitant with HIV infection in 6 subjects and finally after HIV infection diagnosis in 35 subjects.

Demographic, clinical, endoscopic, virological, immunological and serological findings were obtained from the medical records. The following data were collected: age at IBD diagnosis, gender, date of HIV and IBD diagnosis, route of HIV transmission, plasma HIV viral load and CD4 T cell count at IBD diagnosis and relapse, IBD clinical course, IBD treatment, HIV treatment and STIs diagnosis documented in patients’ medical records during the observation period.

HIV infection was diagnosed by ELISA Ag/Ab HIV1/2 and confirmed by a Western blot test. Date of HIV diagnosis was based on the date of the first positive confirmed HIV test. HIV transmission route was self-defined by patient. Syphilis was diagnosed by the Venereal Disease Research Laboratory (VDRL) test, the fluorescent treponemal antibody-absorption (FTA-ABS) and Treponema pallidum hemagglutination assay (TPHA). Neisseria gonorrhoeae or Chlamydia trachomatis infections were initially diagnosed using anamnesis, clinical symptoms and serological findings (IgM and IgG antibodies),and from 2018 using PCR in urine or rectum swab. HAV infection was confirmed serologically by an AbHAV IgM test and HCV infection by HCV RNA detection in serum. STIs were not routinely monitored by gastroenterologists at IBD diagnosis as during IBD relapses. All data was extracted from the patients’ files.

Ethical approval and written informed consent was waived by the Bioethics Committee of the Medical University of Warsaw because of the retrospective nature of the study. Instead of this, Bioethics Committee of Medical University of Warsaw approved the use of oral consent, which was documented in the patients’ medical records. All analyzed patient’s data were fully anonymized. The study followed the principles of the Declaration of Helsinki.

## 3. Results

All studied patients were male with median age 33 years old (range 20–58 years). All, except one, were men who have sex with men (MSM). At the time of IBD diagnosis, 28 patients (56%) had a suppressed HIV viral load (HIV VL < 50 copies/mL) and 12 patients (24%) were detectable, with median HIV VL 51,692 copies/mL (range 499–2,000,000 copies/mL). Their median CD4 cell count was 482 cells/µL (range 165–1073 cells/µL). A total of 21 subjects (42%) were ARV-naive, and 29 subjects (58%) obtained cART along current guidelines. The patients’ characteristics concerning HIV infection status and treatment at the time of IBD diagnosis are presented in Table 1.

All patients, before IBD diagnosis, had typical symptoms such as diarrhea, abdominal and/or rectal pain, rectal bleeding and tenesmus. Endoscopic examinations demonstrated different degrees of inflammatory changes: 28 subjects (56%) had mild degree of observed changes, 19 subjects (38%)—moderate and 3 subjects (6%)—severe. Every patient with severe and 5 patients (16%) with a moderate degree of inflammatory changes needed hospitalization at the time of IBD diagnosis. Endoscopic changes were observed only in the rectum in 30 patients (60%); in the rectum and sigmoid colon in 12 patients (24%); in the rectum, sigmoid and descending colon in 6 patients (12%); and finally in the rectum, sigmoid, descending and transverse colon in 2 patients (4%). Among biopsy findings, CD was diagnosed in 7 patients (14%), UC in 41 patients (82%), and 2 patients (4%) had indeterminate colitis. All studied patients obtained anti-inflammatory drugs after IBD diagnosis, including steroids in 9 cases. Patients treated with steroids had a moderate or severe degree of inflammatory changes in colonoscopy. After the acute phase of IBD, 48 patients received chronically anti-inflammatory drugs. However, 23 of them stopped this treatment—it was the patient or his gastroenterologist decision. A total of 49/50 (98%) patients reported several instances of unprotected receptive anal intercourse and different STIs. Moreover, some of them had an STI diagnosis at the time of IBD confirmation or during the observation period. The complete clinical and endoscopic characteristics of the studied group concerning IBD diagnosis and concomitant STIs are presented in Table 2. All concomitant STIs were treated according to current guidelines.

Only in 10 patients (20%) were one or more IBD relapses observed during the studied period of time. Two of them had CD, the others—UC. Two patients stopped IBD chronic treatment before the first IBD relapse. During the study, 7 patients had only one IBD relapse, 2 patients had two relapses and one patient had seven relapses. All patients with several IBD relapses were on anti-inflammatory treatment. All patients with IBD relapses were successfully treated with ARVs, and their median CD4 cell count was 440 cells/µL (range 217–926 cells/µL). The characteristic of patients with IBD relapses is presented in Table 3.

## 4. Discussion

In our study we present one of the biggest described group of adults diagnosed with IBD and HIV co-existence. Skamnelos et al. searched in PubMed papers concerning concomitant IBD and HIV infection. After excluding several articles due to different reasons, they finally found 13 papers published between 1984 and 2009: 2 case-control studies, 2 case series and 9 case reports. In total, there were 47 patients with IBD and HIV who were included in these studies; the biggest group in a case-control study contained 20 IBD/HIV subjects and in a case series—6 subjects [19]. Small studied groups were the result of the low prevalence of IBD and HIV infection. Recently, a multicenter retrospective cohort study including 65 IBD/HIV patients and 130 without HIV infection and concerning impact of HIV infection on the course of IBD and drug safety profile was published [20]. In our center, until the end of 2019, we have registered 3782 HIV-infected patients; however, we discovered only 50 patients with both diagnoses. Our observation was concomitant with other studies. Viazis et al. found 20 HIV-/IBD-positive patients in their database of more than 1600 individuals with IBD [21]. Sharpstone et al. reported a mean IBD incidence in HIV-positive population as 41 per 100,000 in their 6-year study period [22]. Finally, Yoshida et al. found only 6 patients with HIV and IBD diagnosis among the observed, who comprised 1839 HIV-infected and 1115 IBD-positive individuals [23].

In our study, CD was diagnosed in 7 patients (14%), UC in 41 patients (82%), and 2 patients (4%) had indeterminate colitis. Data concerning CD and UC prevalence in the HIV-positive population are divergent. Landy et al. published the results of a large cohort of HIV-positive and IBD-diagnosed patients treated in London’s Chelsea and Westminster Hospital. In the period of 1999 to 2006, 27 patients had been diagnosed with HIV/IBD, and 19 of them had developed IBD during an already existing HIV infection. The median CD4 cell count at the time of IBD diagnosis was 355 cells/μL. Fifty-three percent (10/19) of the new cases were UC (10/19 cases). The UC incidence in the observed cohort of HIV(+) patients was doubled in comparison to the general population [24]. Conversely, in Viazis et al.’s study, CD was diagnosed in 70% of cases and UC in 30% [21].

Diagnosis of IBD in HIV-infected people is difficult because of several other pathologies that may clinically mimic IBD in this group of patients. Firstly, it could be opportunistic infections such as CMV, HSV, MAC, *Cryptosporidium* spp., *Microsporidium* spp., Isospora belli, Giardia lamblia, Entamoeba histolytica, etc. Secondly, it could be neoplasms, such as gastrointestinal Kaposi’s sarcoma or lymphoma, or several types of STIs. As a result, it is obligatory to confirm IBD diagnosis in histopathological examination to avoid misdiagnosis. All of our patients had endoscopy and IBD confirmation in colorectal biopsy. Recently, Levy et al. showed that, in HIV-infected MSM, some STIs such as Chlamydia trachomatis, lymphogranuloma venereum (LGV), gonorrhea, syphilis and HSV may have similar symptoms and endoscopic findings as IBD [25]. They described 16 patients with colorectal changes misdiagnosed as IBD; 9 of them were initially treated with anti-inflammatory drugs, including 3 patients with steroids. One patient was treated with infliximab. Three rectal swabs were obtained from every subject. As a result, in all studied individuals, the STI diagnosis was confirmed—14 patients were positive for Chlamydia trachomatis, 5 patients for gonorrhea, 4 patients for syphilis, and in 6 patients, several different pathogens were discovered. The STI was diagnosed 1–36 months after the initial IBD diagnosis and its insufficient treatment. Moreover, all studied patients reported unprotected receptive anal intercourse on several occasions. There are other studies confirming these results. Soni et al. showed that, during the proctitis epidemic in the United Kingdom, 12 out of 106 MSM with LGV proctitis were previously misdiagnosed as IBD. The time for correct diagnosis was 2–36 months [26]. Hoie et al. described four MSM (3 of them HIV-positive) with IBD misdiagnosis by gastroenterologists. Their correct diagnosis was LGV infection and was made 9–36 months after IBD diagnosis [27]. Similar results showed Gallegos et al. and Tinmouth et al. [28,29]. Finally, Arnold et al. presented 10 patients with proctocolitis caused by syphilis, LGV or both pathogens in whom pathological findings in endoscopy were very similar to IBD [30]. In our study, all patients except one were MSM and confirmed prior unprotected receptive anal intercourse. This is consistent with the general demographics of HIV in Poland, where MSM aged 30–39 years dominate among HIV-infected individuals [31]. Moreover, a lot of our observed patients had STI diagnosis before or at the time of IBD confirmation or later, during the observation period. Unfortunately, in our center, initially only serological tests for chlamydia and gonorrhea were available to us, and we had limited access to PCR tests for detecting Neisseria gonorheae and Chlamydia trachomatis in rectal swabs until 2018. In 2019, we performed a PCR test for chlamydia and gonorrhea in rectal swabs in all of our HIV/IBD patients to confirm finally their IBD diagnosis, obtaining a positive PCR result in 12 cases. All of the patients with confirmed chlamydia or gonorrhea diagnosis obtained appropriate treatment for their STD with doxycycline, azithromycin, ceftriaxone or penicillin.

Surprisingly, during the observation period, only 10 patients (20%) from our studied group had one or more IBD relapses. Among patients without IBD relapse during 1–18 years of observation, only 2 patients had mild and 8 had moderate or severe IBD degree at the moment of diagnosis. What is more, the majority of them (70%) had initially proctocolitis. The rectosigmoid distribution of disease is also characteristic for several STIs. As a result, we think that about 30% of our HIV/IBD patients without IBD relapse since the moment of IBD diagnosis could have been misdiagnosed and in reality had an STI presenting similarly to IBD. Moreover, in some of them STI may have been an infectious factor inducing IBD. However, our findings and hypotheses need longer observation and confirmation in future studies.

It has been suggested that CD4+ T cells play an important role in IBD pathogenesis. Viazis et al. compared the clinical course of IBD in two groups of patients, IBD/HIV and IBD alone. The majority (14/20) of patients with coexisting HIV/IBD throughout the observation period (median 8.4 years) were immunosuppressed and had a number of CD4 T cells < 500 cells/μL, while almost 70% of them were ARV-treated. In this subgroup of patients, no IBD relapses were observed. Patients with IBD/HIV in comparison to those with IBD alone had significantly fewer disease relapses (relapse rate of 0.016/year in comparison to 0.053/year, respectively). According to the authors, a milder course of IBD in HIV-infected patients could be related to a lower number of CD4+ T cells in serum [21]. However, there are also studies presenting new IBD cases in HIV-infected individuals with CD4 T cell count > 500 cells/μL [23,32]. Moreover, some authors also suggested that the course of IBD in HIV-infected individuals is linked with CD4 T cell function rather than their absolute numbers [33]. Consequently, until now, there has been no final conclusion concerning the role of CD4 T lymphocytes at the first manifestation of IBD. What is more, it has been shown that there are no IBD relapses when the CD4 T cell count is below 200 cells/μL [34]. In our study, we confirmed this finding, as all of our patients with IBD relapses had a CD4 T cell count > 200 cells/μL. Moreover, in patients with several IBD relapses, the CD4 T cell count increased progressively with time and with subsequent IBD deterioration. HIV viral load seems to have no influence on the IBD relapse risk, as all of our patients with IBD exacerbations had an HIV viral load < 50 copies/mL in plasma.

Finally, there are no diagnostic or therapeutic recommendations for patients with HIV/IBD co-infection. European Crohn’s and colitis organization guidelines only recommend HIV testing for every patient with IBD before starting steroid treatment [6]. Additionally, there are no specific guidelines for gastroenterologists to screen for STIs in every proctocolitis case, especially in IBD suspicion. Consequently, according to our personal experience, we strongly recommend HIV and STI testing for every patient, not only MSM, with IBD suspicion and a history of unprotected receptive anal intercourse independently from planned IBD treatment.

Our study has some limitations. Firstly, because of its descriptive and retrospective character, we might miss important data concerning STIs, sexual risk behaviors and other factors as opportunistic infections of gastrointestinal tract influencing the possible misdiagnosis of IBD in HIV-infected subjects. Second, we had no access to the molecular serotyping and genotyping of Chlamydia trachomatis and Neisseria gonorrheae in rectal swab in the moment of IBD diagnosis, and, consequently, it was impossible to make a correct diagnosis in some cases.

## 5. Conclusions

In conclusion, IBD diagnosis in HIV-infected persons is difficult because of several other pathologies, including STIs, that may clinically and histologically mimic IBD in this group of patients. We strongly recommend HIV testing for every MSM patient with IBD suspicion. Moreover, all patients with IBD in the colorectal tract, even confirmed in endoscopy and histopathology, should be tested for STIs using molecular and serological methods. Persons reporting unprotected receptive anal intercourse seem to have the biggest risk of STI-associated proctitis or proctocolitis misdiagnosis and of being treated as IBD. HIV viral load probably has no influence on IBD course in HIV-positive individuals, and the role of peripheral CD4 T cell count in IBD pathogenesis is still controversial. However, we showed that a better immunologic response may favor the higher risk of IBD relapse in individuals on effective cART.

## Figures and Tables

**Table 1 jcm-11-05324-t001:** The patients’ characteristics concerning HIV infection status and treatment at the time of IBD diagnosis.

Case	Route of HIV Infection	Age at IBD Diagnosis (Years)	CD4 Cell Count (Cells/µL)	VL (Copies/mL)	cART
1	MSM	40	627	<40	on treatment
2	MSM	46	637	<40	on treatment
3	MSM	30	621	<40	on treatment
4	MSM	33	458	<40	on treatment
5 ¥	MSM	28	677	NA	naive
6 *	MSM	27	NA	NA	naive
7	MSM	32	1073	<40	on treatment
8 *	MSM	35	NA	NA	naive
9	MSM	31	823	<40	on treatment
10	MSM	30	200	95,000	naive
11	MSM	35	312	<20	on treatment
12	MSM	30	581	51,692	naive
13	MSM	55	441	<40	on treatment
14 *	MSM	27	NA	NA	naive
15	MSM	20	432	<40	on treatment
16	MSM	37	465	<40	on treatment
17	MSM	48	460	<40	on treatment
18	MSM	31	403	<40	on treatment
19	MSM	30	475	55,696	naive
20	IDU	25	490	34,870	naive
21	MSM	28	867	<40	on treatment
22	MSM	31	652	10,050	naive
23	MSM	37	973	<40	on treatment
24 *	MSM	32	NA	NA	naive
25	MSM	37	509	<40	on treatment
26 ¥	MSM	27	243	9400	naive
27	MSM	25	467	<40	on treatment
28	MSM	33	932	11,243	naive
29	MSM	34	454	<40	on treatment
30	MSM	36	721	<40	on treatment
31 ¥	MSM	32	165	67,500	naive
32	MSM	42	841	<40	on treatment
33 *	MSM	40	NA	NA	naive
34	MSM	47	461	315,685	naive
35	MSM	58	321	<40	on treatment
36 *	MSM	45	NA	NA	naive
37	MSM	34	760	<40	on treatment
38	MSM	33	451	<40	on treatment
39	MSM	35	444	<40	on treatment
40 ¥	MSM	22	414	41,000	naive
41	MSM	32	518	<40	on treatment
42	MSM	35	491	<40	on treatment
43 *	MSM	28	NA	NA	naive
44	MSM	24	632	<40	on treatment
45 *	MSM	38	NA	NA	naive
46 ¥	MSM	34	407	71523	naive
47 *	MSM	31	NA	NA	naive
48 ¥	MSM	37	712	546234	naive
49	MSM	29	537	<40	on treatment
50	MSM	25	476	<40	on treatment

MSM—men who have sex with men; IDU—intravenous drug user; VL—HIV viral load; naive—not ARV-treated; cART—combined antiretroviral therapy; on gray background—patients with IBD relapse. * not HIV-infected at IBD diagnosis. ¥ the same date of HIV and IBD diagnosis. On gray background: patients with IBD relapse.

**Table 2 jcm-11-05324-t002:** The clinical and endoscopic characteristics of the studied group concerning IBD diagnosis and concomitant STIs.

Case	Route of HIV Infection	Age in IBD Diagnosis (Years)	HIV Diagnosis (WB Test Year)	IBD Diagnosis (Year)	IBD Diagnosis	Endoscopy ^a^	IBD Degree ^b^	IBD Initial Treatment ^c^	STI during Observation Period ^d^
1	MSM	40	1990	2013	CD	2	MO	5-ASA, steroids	syphilis, acute HCV
2	MSM	46	2007	2013	UC	3	MO	SSZ	syphilis
3	MSM	30	2007	2013	UC	1	MO/H	5-ASA	syphilis, HAV
4	MSM	33	2013	2014	UC	1	MI	SSZ	syphilis, HAV
5 ¥	MSM	28	1996	1996	UC	3	MO	5-ASA	HAV
6 *	MSM	27	1991	1991	UC	3	MO	SSZ, steroids	chronic HCV
7	MSM	32	2008	2013	UC	1	MI	5-ASA	syphilis
8 *	MSM	35	2013	2011	UC	3	MO/H	5-ASA, steroids	syphilis
9	MSM	31	2008	2013	UC	2	MO/H	5-ASA	syphilis, HAV
10	MSM	30	2009	2010	UC	2	S/H	5-ASA, steroids	syphilis
11	MSM	35	2008	2013	indeterminate colitis	1	MI	5-ASA	syphilis
12	MSM	30	2012	2013	UC	1	MO	5-ASA	syphilis, gonorrhea
13	MSM	55	1997	2014	UC	2	MI	5-ASA	none
14 *	MSM	27	2013	2001	UC	4	MI	5-ASA	syphilis
15	MSM	20	2009	2013	UC	2	MO/H	SSZ	gonorrhea, syphilis
16	MSM	37	2008	2015	UC	2	MI	SSZ	syphilis
17	MSM	48	1991	2008	UC	1	MI	5-ASA	gonorrhea
18	MSM	31	2010	2014	UC	2	MO/H	SSZ	syphilis, HAV, acute HCV
19	MSM	30	2012	2013	UC	1	MI	SSZ	syphilis, acute HCV
20	IDU	25	2001	2002	CD	2	MO	SSZ	syphilis
21	MSM	28	2012	2015	UC	1	S/H	5-ASA, steroids	acute HCV, syphilis
22	MSM	31	2013	2014	UC	1	MI	5-ASA	syphilis, acute HCV, HAV
23	MSM	37	2009	2015	UC	1	MI	5-ASA	acute HCV
24 *	MSM	32	2015	2015	UC	2	MO	5-ASA, steroids	syphilis
25	MSM	37	2009	2015	UC	1	MI	5-ASA	syphilis, gonorrhea
26 ¥	MSM	27	2013	2013	CD	3	MO	5-ASA, steroids	HAV, syphilis, acute HCV
27	MSM	25	2012	2013	UC	1	MI	5-ASA	syphilis, HPV
28	MSM	33	2010	2015	indeterminate colitis	1	MI	5-ASA	syphilis
29	MSM	34	2010	2016	UC	1	MI	5-ASA	syphilis, acute HCV
30	MSM	36	2014	2016	UC	1	MI	5-ASA	syphilis, ureaplasma, gonorrhea
31 ¥	MSM	32	2017	2017	UC	1	MI	5-ASA	gonorrhea
32	MSM	42	2005	2017	CD	1	MI	5-ASA	syphilis
33 *	MSM	40	2017	2016	UC	4	S/H	5-ASA, steroids	syphilis, HAV
34 ¥	MSM	47	2017	2018	UC	1	MO	5-ASA	syphilis, HAV
35	MSM	58	2008	2015	UC	1	MI	5-ASA	syphilis
36 *	MSM	45	2018	2017	UC	1	MO	5-ASA, steroids	syphilis
37	MSM	34	2017	2018	UC	1	MI	5-ASA	HAV, syphilis
38	MSM	33	2017	2017	CD	1	MI	5-ASA	HAV
39	MSM	35	2005	2015	UC	1	MI	SSZ	syphilis
40 ¥	MSM	22	2016	2016	UC	1	MI	SSZ	syphilis
41	MSM	32	2017	2019	UC	1	MI	5-ASA	HAV, syphilis
42	MSM	35	2017	2018	UC	1	MI	5-ASA	HAV, Chlamydia
43 *	MSM	28	2018	2017	UC	1	MI	5-ASA	gonorrhea
44	MSM	24	2018	2019	UC	1	MI	5-ASA	Chlamydia, acute HCV
45 *	MSM	38	2018	2016	UC	2	MO	5-ASA	syphilis
46 ¥	MSM	34	2018	2018	CD	3	MO	5-ASA	syphilis, Chlamydia
47 *	MSM	31	2018	2017	UC	2	MO	5-ASA	gonorrhea
48 ¥	MSM	37	2019	2019	UC	1	MI	5-ASA	Chlamydia
49	MSM	29	2018	2019	CD	2	MO	5-ASA	syphilis, gonorrhea, Chlamydia
50	MSM	25	2019	2019	UC	1	MI	5-ASA	syphilis, gonorrhea

MSM—men who have sex with men; IDU—intravenous drug user; IBD—inflammatory bowel disease; STI—sexually transmitted infection; CD—Crohn’s disease; UC—ulcerative colitis; WB—Western blot test; ^a^ 1—endoscopic findings in rectum; 2—endoscopic findings in rectum and sigmoid colon; 3—endoscopic findings in rectum, sigmoid and descending colon; 4—endoscopic findings in rectum, sigmoid, descending and transverse colon. ^b^ MI—mild; MO—moderate; S—severe; H—hospitalization. ^c^ 5-ASA—5-aminosalicylic acid; SSZ—sulfasalazine. ^d^ HCV—hepatitis C virus; HAV—hepatitis A virus. On gray background—patients with IBD relapse. ¥ the same date of HIV and IBD diagnosis. * not HIV-infected at IBD diagnosis. On gray background: patients with IBD relapse.

**Table 3 jcm-11-05324-t003:** The characteristic of patients with IBD relapses.

Case	Age in IBD Diagnosis (Years)	Route of HIV Infection	HIV Diagnosis (WB Year)	IBD Diagnosis (Year)	IBD Diagnosis	Endoscopy ^a^	IBD Degree ^b^	CD4 in IBD Diagnosis (Cells/µL)	VL in IBD Diagnosis (Copies/mL)	Relapses (Year)	CD4 in Relapse (Cells/µL)	VL in Relapse (Copies/mL)
1	40	MSM	1990	2013	CD	2	MO	627	<40	2018	926	<40
2	28	MSM	1996	1996	UC	3	MO	677	NA	2001	235	<50
										2003	447	<50
										2006	320	<50
										2008	548	<50
										2009	440	<50
										2011	463	<50
3 *	27	MSM	1991	1991	UC	3	MO	NA	NA	2000	360	<50
										2008	622	<50
4 *	35	MSM	2013	2011	UC	3	MO/H	NA	NA	2017	217	<40
5	31	MSM	2008	2013	UC	2	MO/H	823	<40	2017	879	<40
6	30	MSM	2009	2010	UC	2	S/H	200	95,000	2015	380	<50
7	48	MSM	1991	2008	UC	1	MI	460	<40	2010	283	<40
8	25	IDU	2001	2002	CD	2	MO	490	34,870	2005	258	<50
										2007	401	<50
9	28	MSM	2012	2015	UC	1	S/H	867	<40	2018	645	<40
10	31	MSM	2013	2014	UC	1	MI	652	10,050	2017	622	<40

MSM—men who have sex with men; IDU—intravenous drug user; IBD—inflammatory bowel disease; CD—Crohn’s disease; UC—colitis ulcerosa; NA—not available; WB—Western blot test. ^a^ 1—endoscopic findings in rectum; 2—endoscopic findings in rectum and sigmoid colon; 3—endoscopic findings in rectum, sigmoid and descending colon; 4—endoscopic findings in rectum, sigmoid, descending and transverse colon. ^b^ MI—mild; MO—moderate; S—severe; H—hospitalization. * IBD before HIV diagnosis.

## Data Availability

The data presented in this study are available on request from the corresponding author.

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
