# Peer review of "Inflammatory Bowel Disease in Adult HIV-Infected Patients-Is Sexually Transmitted Infections Misdiagnosis Possible?"

_jcm, 2022, doi:10.3390/jcm11185324_

Round 1
Reviewer 1 Report (Previous Reviewer 2)
Dear authors,
Thank you for your resubmission.
The manuscript has been greatly improved in the introduction and discussion. The results have been detailed for this study. However, to my personal opinion of MD and Postdoc PhD (being recently under the rule of a rigorous PhD advisor), the provided extensive modifications of this paper might need another round of modifications in order to ease reading and to improve the scientific rationale of this interesting paper.
I have a few minor arrangements (English writing) and formatting and scientific content :
line 13 : "were confirmed BY biopsy".
line 14 : Results : (two dots, not one)
line 14 : "with a median age of"
line 17 : "Patients had no CD no UC diagnosis" : This is generally designated as "Indeterminate colitis".
line 20-21 : Please rephrase correctly : Should testing be performed in IBD patients with colorectal inflammation, or should colorectal testing be performed in IBD patients ? Are there particular forms of IBD which could predispose to HIV ? Ileitis ? Colitis ? Rectitis ?
line 31 : Are you sure you have rectal bleeding in an ileal form of Crohn's disease ? Is it "usually" ?
line 37 ; please rephrase, for example "CD potentially affects any part of the gastrointestinal tract, while UC only affects the colon and rectum".
line 36 : The statement "more destructive" is false. There is way more destruction in UC than CD. I would tend to describe UC as "extended hemorragic ulcerations of the rectal and colonic mucosae" and CD as "transmural inflammation potentially affecting the intestinal tract, creeping fat, and sometimes epithelioid granulomas". However, CD patients actually do have more risk of strictures and fistulae and peritonitis.
line 39 : "The etiopathogenesis of IBD" or "The etiology of IBD" or "The causative mechanisms of IBD". I Don't think "the IBD primary cause" sounds right in English langage.
line 43 : Do the authors mean "persistent" or "preserved" ? I'm guessing persistent inflammation. I would personally rephrase the whole sentence by saying "One of the major hypotheses which could explain IBD etiopathogenesis is persistent inflammation triggered by an unkown environmental antigen […]".
line 46 47 : "GALT is the major lymphoid tissue in the body". I would personally remove that sentence or provide appropriate references.
lines 49-87 : I would like to congratulate the authors for this very interesting paragraph, Thank you
Pages 3 and 4 : I would tend to reassemble the table in one page for easier reading (If possible).
Same modifications for Table 2 in pages 6 and 7.
Same modifications for Table 3 in pages 7 and 8.
Author Response
line 13 : "were confirmed BY biopsy". - we changed this phrase among suggestion
line 14 : Results : (two dots, not one) - we left one dot to standardize the style of abstract
line 14 : "with a median age of" - we introduced suggested changes
line 17 : "Patients had no CD no UC diagnosis" : This is generally designated as "Indeterminate colitis". - we used 'indeterminate colitis' in all parts of the manuscript, also in Table 2
line 20-21 : Please rephrase correctly : Should testing be performed in IBD patients with colorectal inflammation, or should colorectal testing be performed in IBD patients ? Are there particular forms of IBD which could predispose to HIV ? Ileitis ? Colitis ? Rectitis ? - we modified this phrase among suggestion
line 31 : Are you sure you have rectal bleeding in an ileal form of Crohn's disease ? Is it "usually" ? - we removed 'rectal bleeding' and 'usually' from this phrase
line 37 ; please rephrase, for example "CD potentially affects any part of the gastrointestinal tract, while UC only affects the colon and rectum". - we modified this sentence among suggestion
line 36 : The statement "more destructive" is false. There is way more destruction in UC than CD. I would tend to describe UC as "extended hemorragic ulcerations of the rectal and colonic mucosae" and CD as "transmural inflammation potentially affecting the intestinal tract, creeping fat, and sometimes epithelioid granulomas". However, CD patients actually do have more risk of strictures and fistulae and peritonitis. - we modified this sentence among suggestion
line 39 : "The etiopathogenesis of IBD" or "The etiology of IBD" or "The causative mechanisms of IBD". I Don't think "the IBD primary cause" sounds right in English langage. - we used 'the causative mechanisms of IBD'
line 43 : Do the authors mean "persistent" or "preserved" ? I'm guessing persistent inflammation. I would personally rephrase the whole sentence by saying "One of the major hypotheses which could explain IBD etiopathogenesis is persistent inflammation triggered by an unkown environmental antigen […]". - we changed this phrase among suggestion
line 46 47 : "GALT is the major lymphoid tissue in the body". I would personally remove that sentence or provide appropriate references. - we removed this sentence from the text
lines 49-87 : I would like to congratulate the authors for this very interesting paragraph, Thank you - thank you very much for your appreciation
Pages 3 and 4 : I would tend to reassemble the table in one page for easier reading (If possible). - it is impossible because Table 1 is too big
Same modifications for Table 2 in pages 6 and 7. - we modified Table2
Same modifications for Table 3 in pages 7 and 8. - we left Table 3 unchanged because this Table is located on one page
Finally, according to suggestions of both Reviewers, our manuscript had been checked by an English-speaking colleague (all changes we highlighted in yellow).
Reviewer 2 Report (Previous Reviewer 1)
I am happy that the revised publication is suitable for publication.
Author Response
Thank you for your appreciation.
According to suggestions of both Reviewers, our manuscript had been checked by an English-speaking colleague (all changes we highlighted in yellow).
Best regards.
This manuscript is a resubmission of an earlier submission. The following is a list of the peer review reports and author responses from that submission.
Round 1
Reviewer 1 Report
Thanks for the opportunity to review this interesting retrospective study of IBD in HIV positive patients (almost exclusively MSM). The authors present data making a compelling case that many of these patients may be misdiagnosed with IBD and may actually have STIs.
Some suggestions for improvement follow:
1. The methods are inadequately described. What was the case definition for IBD? How were such cases located? Was this a study involving interrogating old medical files? If so what search was done and how? What time point were data obtained from? How was followup data obtained (to confirm/ deny relapse with IBD) and how likely is this data to be complete (ie was the patient population captive)? How complete was STD testing at the time of diagnosis of IBD? Major improvements need to be made to this section such that the study methodology is laid out in sufficient detail that a reader could attempt to replicate the methods. I suggest looking at the STROBE checklist help with this (Checklists - STROBE (strobe-statement.org)
2. There is multiple sources of evidence within the data obtained that point to the IBD diagnosis being wrong, with the authors only discussing some of these. They include:
a. The patient cohort being almost all MSM. Probably this is different from the wider demographics of HIV in Poland? The authors should discuss this as it points to the presence of a confounder (ie anal intercourse increases both the risk of HIV and perianal infection, which acts as a confounder for apparent IBD, leading to a greatly increased apparent IBD rate in this group.)
b. The distribution of disease being very predominantly rectosigmoid - where STD proctitis occurs.
c. The lack of recurrence among this group of patients with IBD.
There may be other reasons as well, but it appears based upon these arguments and the data presented that the majority of patients are likely to have had a STD and not IBD.
Overall then the study will be a good addition to the literature once the above issues are addressed.
Author Response
To Whom It May Concern,
My Colleagues and I would like to present to you a revised manuscript regarding a study we conducted on possible misdiagnosis of sexually transmitted infections and inflammatory bowel disease in adult HIV-infected patients for your consideration. We submitted the previous version of the manuscript on 29th May 2022 (manuscript id: jcm-1769691) and received a number of very helpful and astute comments from the reviewers. Finally, the overall Editor’s decision was to make major manuscript revision with a possibility of resubmission.
After careful study of the reviewers’ pointers and extensive discussion among the authors of the manuscript, we revised large parts of the manuscript (which we highlighted for convenience). We would like to thank the Editors of Journal of Clinical Medicine and the reviewers of the previous manuscript for your invaluable contribution to this manuscript.
As this is a resubmission, we would like to use the remainder of this cover letter to address the concerns of the reviewers of the previous version of the manuscript and to highlight the steps we took to alleviate them in the manuscript.
Both the reviewers were of the opinion that Introduction, Methods and Results sections should be improved. After discussion we agree that they should be rewritten. In Introduction section, among Reviewer’s suggestion, we added large fragments about IBD definition and diagnosis, IBD physiology, co-existence of IBD and STIs as co-existence of IBD and HIV. Moreover, we added nine new references for Introduction section. In Methods section, among Reviewer’s suggestion, we described IBD case definition, how every IBD case was located, the methodology of collecting all data from the medical records and we preceded STI testing methods till 2018 and after 2018. In Result section, we modified Table 1 and Table 2 for more clarity.
One of the reviewers pointed out that the patients cohort being almost MSM and that this is probably different from the wider demographics of HIV in Poland as it could be a confounder. According to all epidemiological data the majority of new HIV diagnoses in Poland are among MSM. This is reflected in MSM distribution in our center, where roughly 90% of patients are men, especially MSM. However, we added this information in Discussion section.
One of the reviewers suggested that robust statistical analysis and detailed microbiological analysis could improve the quality of our paper. However, our study was a retrospective descriptive study. Our population group contains only 50 persons. We believe that this is too small group for making multivariate analysis. Though, we will consider this idea for the next paper comparing HIV-infected patients with IBD with HIV-negative patients with IBD. In Methods section, we also described all STIs diagnostic tests which we used during observation period.
We confirm that neither the manuscript nor any parts of its content are currently under consideration or published in another journal.
All authors have approved the manuscript and agree with its submission to Journal of Clinical Medicine.
We certify that the paper is prepared according to the 'Instructions for Authors'.
We look forward to your reply.
Yours faithfully,
Magdalena M. Suchacz and Colleagues
Department of Infectious and Tropical Diseases and Hepatology
Medical University of Warsaw, Poland
Reviewer 2 Report
Hello,
To all authors, thank you for the interesting work. IBD and STDs are generally two events that can occur in IBD patients although not so common in general practice, except in the case of working in a specialized IBD center.
I have minor comments :
When conducting an epidemiological study about pathogens and IBD, I would tend from my reading to add precisions to serological evidence and serological values (Antibody titers ? IgM ? IgG ?). I would also advise to be able to know Ct values for bacteriological and virological analyses.
STD diagnosis could be more precise. "Syphilis before and after IBD" is somewhat unclear and doesn't explain the timelapse of infection and the appearance of gastrointestinal symptoms. Same for "just before IBD".
From a statistical point of view, it seems like there are multiple parameters such as IBD, HIV, IBD treatment, HIV treatment, co infections by STDs ... These parameters seem to fit to a multivariate analysis. Patients should be paired by sex, age, ethnicity, etc ... multiple parameters should be taken into count in this complex question.
I would tend to expect more in the Introduction about STDs and IBD, HIV and IBD, and also about the physiopathology of IBD.
Globally, from my point of view, this is a good paper with interesting ideas; however robust statistical analysis, detailed microbiological analysis and more explanation about the subject could really improve the quality of this paper. Also presentation (Tables overlapping in different pages) could be improved for more clarity.
Best regards
Author Response

(The authors gave the same response as above.)
